# Analytical Estimation of Data-Motivated Time-Dependent Disease Transmission Rate: An Application to Ebola and Selected Public Health Problems

**DOI:** 10.3390/tropicalmed6030141

**Published:** 2021-07-31

**Authors:** Anuj Mubayi, Abhishek Pandey, Christine Brasic, Anamika Mubayi, Parijat Ghosh, Aditi Ghosh

**Affiliations:** 1PRECISIONheor, Los Angeles, CA 90025, USA; amubayi@ilstu.edu; 2Department of Mathematics, Illinois State University, Normal, IL 61790, USA; 3Center for Collaborative Studies in Mathematical Biology, Illinois State University, Normal, IL 61790, USA; 4College of Health Solutions, Arizona State University, Tempe, AZ 28608, USA; 5Yale School of Public Health, Yale University, New Haven, CT 06520, USA; abhishek.pandey@yale.edu; 6Department of Mathematics and Statistics, University of Wisconsin-Whitewater, Whitewater, WI 53190, USA; brasiccs23@uww.edu; 7Department of Chemistry, University of Allahabad, Allahabad 211002, India; anamika.mubayi@gmail.com; 8School of Medicine-Family and Community Medicine, University of Missouri, Columbia, MO 65201, USA; phool.ghosh@gmail.com; 9Department of Mathematics, Texas A & M-Commerce, Commerce, TX 75429, USA

**Keywords:** transmission coefficient, infectious disease dynamics, compartmental model, parameter estimation, epidemic modeling

## Abstract

Obtaining reasonable estimates for transmission rates from observed data is a challenge when using mathematical models to study the dynamics of ?infectious? diseases, like Ebola. Most models assume the transmission rate of a contagion either does not vary over time or change in a fixed pre-determined adhoc ways. However, these rates do vary during an outbreak due to multitude of factors such as environmental conditions, social behaviors, and public-health interventions deployed to control the disease, which are in-part guided by changing size of an outbreak. We derive analytical estimates of time-dependent transmission rate for an epidemic in terms of either incidence or prevalence using a standard mathematical SIR-type epidemic model. We illustrate applicability of our method by applying data on various public health problems, including infectious diseases (Ebola, SARS, and Leishmaniasis) and social issues (obesity and alcohol drinking) to compute transmission rates over time. We show that time-dependent transmission rate estimates can have a large variation, depending on the type of available data and other epidemiological parameters. Time-dependent estimation of transmission rates captures the dynamics of the problem better and can be utilized to understand disease progression more accurately.

## 1. Introduction

An epidemic is a function of environmental factors and a contact structure that varies over time, which in turn leads to varying transmission potential of an “infection”. We also refer the word “infection” to describe social influences exerted by a typical influential individual with a particular social problem that results in a naive (to the social problem) individual becoming involved in the problem. For example, an alcohol drinker might influence an abstainer into imitating drinking behavior and initiating alcohol drinking. Many authors have studied outbreaks of social problems and infectious diseases using compartmental transmission/influence model. Qualitative aspects of homogeneous mixing models with constant transmission potential of an infection are well understood for various applications. These models are relatively easy to analyze and can answer questions, at the population level, with good precision. Homogeneous mixing compartmental models have a long history; however, quantification of temporal transmission potential of an infectious agent, an input variable for this type of model, has been a challenge.

William Hamer first published a paper in 1906 containing an epidemic model for the transmission of measles where his observation included the incidence of new cases, in a time interval, is proportional to the product, SI, of the density of susceptibles (*S*) and the density I of infectives (*I*) in the population. The formulation of incidence can be explained by considering some epidemiological quantity. Consider a single susceptible individual in a homogeneously mixing population of size *N*. This individual contacts other members of the population at the rate *c*, per unit time, and a proportion I/N of these contacts are with individuals who are “infectious”. If the probability of transmission of infection given contact is ρ, then the rate at which the infection is transmitted to a susceptible is ρcI/N, per unit time, and the rate at which the susceptible population becomes infected is ρcSI/N.

The contact rate is often a function of population density, reflecting the fact that contacts take time and saturation occurs. If *c* is assumed approximately proportional to *N* or equal to constant, incidence can be represented by terms such as βSI (referred as *mass action incidence*) or βSI/N (referred as *standard incidence*), respectively. The parameter β, which includes the contact rate *c*, is known as a “transmission coefficient” (or “effective contact rate” or “transmission potential”) with units as ${\mathrm{time}}^{-1}$. At low population densities mass action is a reasonable approximation of a much more complex contact structure; however, in general, standard incidence is more appropriate for modeling transmission for human diseases or influences for social problems. The term βI/N is sometimes referred as the force of infection, i.e., per-capita rate at which susceptible members of the host population are becoming infected. On the other hand, the transmission rate, represents the number of new infections per unit of time generated by an infected individual. The transmission rate is calculated by dividing incidence for a given time period by a disease prevalence for the same time interval.

Most infectious disease data are collected in form of incidence and/or prevalence. *Prevalence* of a “disease” in a population is defined as the total number of cases of the disease in the population at a given time, whereas *prevalence proportion* is computed by dividing the total number of cases in the population by the number of individuals in the population. It is used as an estimate of how common a condition is within a population over a certain period of time. *Incidence* is a measure of the risk of developing some new condition within a specified period of time. Incidence proportion (also known as cumulative incidence) is the number of new cases within a specified time period divided by the size of the population initially at risk. When the denominator is the sum of the person-time of the at-risk population, it is also known as the incidence density rate or person-time incidence rate. Using person-time rather than just time handles situations where the amount of observation time differs between people, or when the population at risk varies with time. Prevalence is a measurement of all individuals affected by the disease within a particular period of time, whereas incidence is a measurement of the number of new individuals who contract a disease during a particular period of time. So, prevalence and incidence proportion at the time *t* is given by I(t)/N(t) and β×(S(t)/N(t))×(I(t)/N(t)), respectively.

In compartmental mathematical models, varied assumptions are made based on characteristics of a modeling disease which lead modelers to focus on more important aspects of the epidemic. For example, an epidemic that occurs on a timescale that is much shorter than that of the population replenishment (that is, epidemic occurs at a much faster rate than births and deaths in the population), constant population size can be assumed. Additional common features of these models might include temporary or permanent recovery of infected individuals and a birth rate into infective class. Whether establishment or a major outbreak of an infectious disease or a social problem will occur in a population, requires extensive experience or a mathematical model of disease dynamics and estimates of the parameters of the disease model. Here, we provide a method for estimating the transmission coefficient (β), which is a key parameter in shaping the epidemic dynamics generated from the model because of nonlinearity associated with the term containing it. A suitable set of data for estimation of β includes prevalence and incidence of the outbreak in question. There are many different methods for estimating β but most of them results in an aggregate value over time. The methods in the literature include estimation using regression of prevalence and time since start of an epidemic [1], estimating from equation for basic reproductive rate when threshold density is known [2], estimating from equilibrium prevalence [3,4], using age prevalence curves [5], inferring from behavior or contact data [6], and iterative comparison of field prevalence data with model predictions [7].

Some researchers have modeled time-varying transmission coefficients for diseases that follow seasonal patterns but using a predefined functional form [8]. On the other hand, a study by Finkenstadt and Grenfell [9] uses a discrete time model that allows for a temporally varying transmission parameter with a period of one year with no assumption on functional form. However, their estimation is computationally intensive and assumes that reporting interval of the available data must be an integer fraction of the serial interval of the disease. Another study by Pollicott et al. (2012), suggested first to fit the data with a pre-defined continuous function and then provide an analytical estimate of the transmission rate. However, their method was only applicable to prevalence data, with some restrictive assumptions on the initial number of susceptible or vital rates [10]. In the current study, we provide an analytical estimation of transmission coefficient using distinct and novel mathematical approach that is not only applicable to both prevalence and incidence data but also has its applicability to wide public-health problems including social issues. Table 1 provides a brief comparison of the estimation procedures in the Pollicott et al and the present study.

Examples of social problems such as alcohol drinking and obesity and infectious diseases such as Ebola, Visceral Leishmaniasis (or Kala-azar), and SARS are used to show relevance of the analytical work. The available data of US college alcohol drinking and obesity outbreak in US include prevalence trends, whereas incidence data of Ebola outbreak in West Africa (Guinea, Sierra Leone, and Liberia), Kala-azar outbreak in Bihar, and SARS epidemic in Hong Kong are used for the estimation.

In this paper, we compute time-dependent and -independent transmission coefficient of Ebola virus disease along with other health care problems such as college alcohol drinking, the obesity epidemic in United States, the spread of Visceral Leshmaniasis, and the spread of the 2003 SARS Outbreak in Hong Kong. The remaining paper is stratified as follows: Section 2 provides a compartmental SIR model and two analytical expressions of transmission coefficients based on prevalence and incidence data; examples for computing coefficient over time using each of the two expressions and field data are shown in Section 3; and finally, the results are discussed in Section 4. Figure 1 represents the overview of this paper.

## 2. Materials and Methods

### 2.1. Formulation for Time-Dependent Estimation

Epidemics in a population are typically captured via an SIR-type (Susceptible-Infectious- Recovered) epidemic models. It is assumed that in a well-mixed population, individuals interact with each other at random. The model considers a population of size *N*, where S(t), I(t) and R(t) represents number of susceptible, infectious, and removed individuals at time *t*. Individuals are recruited in the population at the rate b(t), die at the μ constant per-capita mortality rate and recover from infection at a α constant per-capita rate. The model assumes that the recruitment rate is governed by immigration, emigration and natural births, and the recovered individuals are immune to the infection but after temporary immunity period a recovered individual may lose immunity and move to *S* class. Hence, a “disease” outbreak in population can be captured by the following system of differential equations: (1)$$\begin{array}{cc} \frac{dS}{dt} & =pb(t)-\beta (t)SI+\gamma (t)R-\mu (t)S \end{array}$$(2)$$\begin{array}{cc} \frac{dI}{dt} & =\bar{p}b\left(t\right)+\beta \left(t\right)SI-\alpha \left(t\right)I-\mu \left(t\right)I \end{array}$$
where R(t)=1−S(t)−I(t) and parameters are defined in Table 2 and Table 3, Figure 2.

Following steps carried out in Hadeler [11] and using Equations (Equation 1) and (Equation 2), we derive two explicit expressions for β(t): one based on prevalence data and other on the incidence of the disease. The main derivation steps for are mentioned below.

#### 2.1.1. Derivation of β(t) Expression in Terms of Prevalence

Suppose prevalence data are available. Derivation of β(t) as a function of prevalence is carried out as follows. Adding Equations (Equation 1) and (Equation 2) we obtain
(3)$${\left(S+I\right)}^{\prime }=\left(b\left(t\right)+\gamma \left(t\right)\right)-\left(\gamma \left(t\right)+\mu \left(t\right)\right)\left(S+I\right)-\alpha \left(t\right)I$$

Setting c(t)=b(t)+γ(t) and d(t)=γ(t)+μ(t) in Equation (Equation 3) and solving it we obtain
(4)$$S\left(t\right)=\left(S\left(0\right)+I\left(0\right)\right)Z\left(0,t\right)+\int_{0}^{t}Z\left(s,t\right)\left[c(s)-\alpha (s)I(s)\right]ds-I\left(t\right)$$
where $Z\left(a,b\right)=exp\left(-\int_{a}^{b}d\left(s\right)ds\right)$.

Isolating β(t) from Equation (Equation 2) we obtain β(t) as function of prevalence (*I*)
(5)$$\beta \left(t\right)=\frac{I^{\prime }+\left(\alpha \left(t\right)+\mu \left(t\right)\right)I-\bar{p}b\left(t\right)}{SI}$$
where S(t) is given by Equation (Equation 4).

Note, beside prevalence (*I*), we also need $I^{\prime }$ to compute β(t) using Formula (Equation 5). However, $I^{\prime }$ can be approximated using prevalence data.

#### 2.1.2. Derivation of β(t) Expression in Terms of Incidence

On the other hand, suppose incidence data are available. To calculate expression of β(t) as a function of incidence (w(t)=β(t)SI) we first solve Equation (Equation 2) for *I* with initial condition I(T) (where T∈[0,L] is a time at which the prevalence proportion, I(T), is available) and obtain
(6)$$I\left(t\right)=I\left(T\right)H\left(T,t\right)-\int_{t}^{T}H\left(s,t\right)\left[w\left(s\right)+\bar{p}b\left(s\right)\right]ds$$
where $H\left(a,b\right)=exp\left(-\int_{a}^{b}\left(\alpha \left(s\right)+\mu \left(s\right)\right)ds\right)$.

Using this expression of I(t) in Equation (Equation 1) and solving the resultant equation for *S* with initial condition S(0) we obtain
(7)$$\begin{array}{cc} S(t)= & Z\left(0,t\right)S\left(0\right)+\int_{0}^{t}Z\left(u,t\right)\left[pb(u)-w(u)\right]du\\ \; & +\int_{0}^{t}Z\left(u,t\right)\gamma \left(u\right)\left[1-I\left(T\right)H\left(T,u\right)+\int_{u}^{T}H\left(s,u\right)\left[w\left(s\right)+\bar{p}b\left(s\right)\right]ds\right]du \end{array}$$

Thus,
(8)$$\beta \left(t\right)=\frac{w(t)}{SI}$$
where S(t) and I(t) are given by Equations (Equation 6) and (Equation 7), respectively.

Note, we need prevalence at time point T, I(T), to compute β(t) using Formula (Equation 7). The time point *T* can be appropriately chosen, close to maximum of prevalence and not towards starting or end of epidemic.

### 2.2. Time-Independent Estimation: Bayesian Analysis

The Bayesian Monte Carlo Markov Chains (MCMC) approach can be used to quantify uncertainty around the transmission rates and compare our analytical estimates with it.

Let θ represents vector of our transmission parameters and $y={\left(y_{1},y_{2},\ldots \ldots ,y_{T}\right)}^{T}$ is the available data set. We can take likelihood function in our Bayesian approach as
(9)$$L\left(y|\theta \right)=\prod_{i=1}^{T}\left(\frac{1}{\sqrt{2\pi \sigma^{2}}}\right)\times \exp\left(-\frac{{\left[\;logit\left(y_{i}\right)-logit\left(f_{i}\left(\theta \right)\right)\;\right]}^{2}}{2\sigma^{2}}\right)$$
where *T* is the total number of data points in the data set, σ is the appropriately chosen variance and f(θ) is the model output function for which data are used. If more than one data sets are used then the likelihood can be modified as follows:$$L\left(y|\theta \right)=\prod_{k}L\left(y_{k}|\theta \right)$$

Although a Bayesian approach can provide uncertainty around time-independent average transmission rate, it does not inform how the transmission rate varied over time and uncertainty itself is constant over time. Therefore, this approach, while assists in understanding uncertainty in disease progression, it does not address the challenge of capturing changing transmission rates over the progression of an epidemic with respect to time.

## 3. Results

We use four examples to show how to estimate β over time from the available epidemiological data. The examples provide a method to study social and public-health issues. To compute estimates of β(t), we use first order discretization for derivatives and composite trapezoidal rule for integration as given below
$$f^{\prime }\left(t\right)\approx \frac{f(t+h)-f(t)}{h}$$
$$\int_{0}^{t}f\left(x\right)dx\approx h\left(\frac{f(0)+f(t)}{2}+\sum_{k=1}^{n-1}f\left(kh\right)\right).$$

These discretizations are used in the formulas given in Equations (Equation 5) and (Equation 8).

We can avoid this discretization by choosing a function, for example, a polynomial that can be fitted to the prevalence and incidence temporal data. This fitted function can then be used directly in Equations (Equation 5) and (Equation 8). Additional demographical and epidemiological data that we require in the β(t) estimation for both incidence and prevalence case are duration of infectious period, recruitment rate, natural mortality rate, and relapse rate.

### 3.1. Using Incidence Data

In this section, we apply available incidence data to three past epidemics: the 2014–2016 Ebola outbreak in West Africa, the 2005 outbreak Visceral Leishmaniasis in the Indian state of Bihar, and the 2003 SARS outbreak in Hong Kong.

#### 3.1.1. 2014–2016 Ebola Outbreak in West Africa

In this section, we estimate the transmission coefficient, β(t) for the 2014–2016 Ebola epidemic in West Africa using available incidence data. The number of reported cases per month were retrieved from the Center for Disease Control and Prevention (CDC) and are shown totaled as West Africa (Figure 3a), and individually for Guinea (Figure 3c), Sierra Leone (Figure 3e), and Liberia (Figure 3g) [21]. For these estimates, prevalence is taken as 31 May 2015, as this point is close to the maximum prevalence and not towards the start of the epidemic (see Section 2.1.2). Incidence is calculated by dividing these case counts by the 2016 population for each country, as reported by the United Nations (UN) [22]. We assume a constant recovery rate of 10 days (α(t)=α), a constant relapse rate of 10 years (γ(t)=γ), no vertical transmission (p=1), and a constant population (b(t)=μ(t)=u=0); since the CDC data provides monthly case counts, these parameters are adjusted to per month rates. We estimate β(t) by simplifying Equation (Equation 6) as follows:(10)$$\begin{array}{c} \beta \left(t\right)=\frac{w(t)}{\left(S\left(0\right)-\int_{0}^{t}w\left(u\right)du\right)\left(I\left(T\right)e^{\alpha (T-t)}-\int_{t}^{T}e^{\alpha (s-t)}w\left(s\right)ds\right)} \end{array}$$

On discretizing Equation (Equation 10) we obtain the following expressions. If t≤T,
(11)$$\beta \left(t\right)\approx \frac{w(t)}{a_{1}b_{1}}$$
where
$$a_{1}=S\left(0\right)-h\left[\frac{w(0)+w(t)}{2}+\sum_{k=1}^{n-1}w\left(kh\right)\right]$$
and
$$b_{1}=I\left(T\right)e^{\alpha (T-t)}-he^{-t\alpha }\left[\frac{g_{1}\left(t\right)+g_{1}\left(T\right)}{2}+\sum_{k=1}^{n-1}g_{1}\left(kh\right)\right]$$

If t>T,
(12)$$\beta \left(t\right)\approx \frac{w(t)}{a_{1}b_{2}}$$
where
$$b_{2}=I\left(T\right)e^{\alpha (T-t)}+he^{-t\alpha }\left[\frac{g_{1}\left(t\right)+g_{1}\left(T\right)}{2}+\sum_{k=1}^{n-1}g_{1}\left(kh\right)\right]$$
where $g_{1}\left(x\right)=e^{m_{1}x}w\left(x\right)$ and $m_{1}=\alpha$.

For the estimation of β(t) with regards to available incidence data, the estimates are found in Table A1 (see Appendix A) and are shown for West Africa (Figure 3b), Guinea (Figure 3d), Sierra Leone (Figure 3f), and Liberia (Figure 3h). Comparing the results for each region, we find the largest temporal estimate for both the mean and median β(t) to be that of Guinea (see Table 4 and Figure 4). Analyzing the estimates for transmission rate temporally, we observe that transmission rate follows the incidence pattern reflecting the effects of exponential incline in the beginning of epidemic as well as impacts of disease-acquired immunity as well as non-pharmaceutical interventions implemented over the course of epidemic (Figure 3).

#### 3.1.2. 2005 Occurrence of Visceral Leishmaniasis in Bihar, India

Visceral Leishmaniasis (VL) is a vector borne infectious disease that is spread from person to person by a bite of the tiny insect, sandfly. Large population suffers from VL in some tropical and subtropical countries of the world. The highest burden of the VL is found in Indian state of Bihar. We obtained underreporting adjusted 2005 incidence data of Bihar from [7]. The data contain number of new cases during past month adjusted for underreporting. The Expression (Equation 13) is used to estimate β(t) via two different models. The first model was for a single outbreak and hence demography was not considered whereas the second model assumed birth and death though with a same per-capita rate.

If t≤T then
(13)$$\beta \left(t\right)\approx \frac{w(t)}{a_{2}b_{3}}$$
where
$$a_{2}=1-e^{-\mu t}\left(1-S\left(0\right)\right)-he^{-tm_{2}}\left[\frac{g_{2}\left(0\right)+g_{2}\left(t\right)}{2}+\sum_{k=1}^{n-1}g_{2}\left(kh\right)\right]$$
and
$$b_{3}=I\left(T\right)e^{\left(\alpha +\mu )(T-t\right)}-he^{-tm_{3}}\left[\frac{g_{3}\left(t\right)+g_{3}\left(T\right)}{2}+\sum_{k=1}^{n-1}g_{3}\left(kh\right)\right]$$

If t>T
(14)$$\beta \left(t\right)\approx \frac{w(t)}{a_{2}b_{4}}$$
where
$$a_{3}=1-e^{-\mu t}\left(1-S\left(0\right)\right)-he^{-tm_{2}}\left[\frac{g_{2}\left(0\right)+g_{2}\left(t\right)}{2}+\sum_{k=1}^{n-1}g_{2}\left(kh\right)\right]$$
and
$$b_{4}=I\left(T\right)e^{\left(\alpha +\mu )(T-t\right)}+he^{-tm_{3}}\left[\frac{g_{3}\left(t\right)+g_{3}\left(T\right)}{2}+\sum_{k=1}^{n-1}g_{3}\left(kh\right)\right]$$
where $g_{i}\left(x\right)=e^{m_{i}x}w\left(x\right)$ (for i=2,3), $m_{2}=\mu$ and $m_{3}=\alpha +\mu$.

Since annual epidemic during 2005 started showing clear trend of decaying in the month of October, we took this time to compute the prevalence of VL in Bihar. Prevalence during October 2005 was computed under assumption that 25% of worldwide leishmaniasis prevalence is from VL cases whereas remaining is from other forms of Leishmaniasis. It also assumed 20% of global burden is in Bihar. Since some proportion of a population are naturally immune to the disease, we carried out estimation for three different values of initial proportion of susceptibles, namely 0.1, 0.5 and 0.8. Recovery rate of 0.21 per month and influx/outflux rate of the population of 0.00138 was computed using data from Mubayi et al. (2010) [7]. The other assumptions of the model include constant recovery (i.e., α(t)=α), no vertical transmission (i.e., p=1), permanent recovery (i.e., γ(t)=0) and same constant per-capita incoming and outgoing rates (i.e., b(t)=μ(t)=μ). We only model human population and do not take into account vector population explicitly. Thus, β(t) could be interpreted as vectorial capacity of sandfly population transmitting infection between humans.

The obtained estimates of β(t) are given in Table A2 (see Appendix A) and Figure 5a,b and Figure 6a,b. The β estimates that we have computed here are comparable to corresponding estimates in [7] (in this reference the mean estimates are $\beta_{h}=0.13$ (with median = 0.11, SD = 0.08, Q1 (25th percentile) = 0.07, Q3 (75th percentile) = 0.17) and $\beta_{v}=0.12$ (with Median = 0.11, Std = 0.08, Q1 (25th percentile) = 0.07, Q3 (75th percentile) = 0.16) where around 75% of the population was susceptible).

#### 3.1.3. 2003 SARS Outbreak in Hong Kong

Severe acute respiratory syndrome (SARS) is a viral respiratory illness caused by a coronavirus. SARS epidemic in Hong Kong is shown in Figure 7a. We estimated transmission coefficient using a single outbreak model with parameters values given in Table A3. The formula used for estimating β(t) is
(15)$$\beta \left(t\right)=\frac{w(t)}{\left(S\left(0\right)-\int_{0}^{t}w\left(u\right)du\right)\left(I\left(T\right)e^{\alpha (T-t)}-\int_{t}^{T}e^{\alpha (s-t)}w\left(s\right)ds\right)}$$

On discretizing Equation (Equation 15) we obtain following expressions. If t≤T,
(16)$$\beta \left(t\right)\approx \frac{w(t)}{a_{4}b_{5}}$$
where
$$a_{4}=S\left(0\right)-h\left[\frac{w(0)+w(t)}{2}+\sum_{k=1}^{n-1}w\left(kh\right)\right]$$
and
(17)$$b_{5}=I\left(T\right)e^{\alpha (T-t)}-he^{-t\alpha }\left[\frac{g_{4}\left(t\right)+g_{4}\left(T\right)}{2}+\sum_{k=1}^{n-1}g_{4}\left(kh\right)\right]$$

If t>T,
(18)$$\beta \left(t\right)\approx \frac{w(t)}{a_{4}b_{6}}$$
where
(19)$$b_{6}=I\left(T\right)e^{\alpha (T-t)}+he^{-t\alpha }\left[\frac{g_{4}\left(t\right)+g_{4}\left(T\right)}{2}+\sum_{k=1}^{n-1}g_{4}\left(kh\right)\right]$$
where $g_{4}\left(x\right)=e^{m_{4}x}w\left(x\right)$ and $m_{4}=\alpha$.

The temporal estimates of β(t) are shown in Table A3 (see Appendix A) and Figure 7b.

### 3.2. Using Prevalence Data

We use US national college alcohol drinking and obesity data as examples in this section. In Appendix A, We also present a hypothetical example with synthetic prevalence data and known time-varying transmission rate to illustrate the ability of our analytical expression to accurately capture the time-dependent transmission rate.

#### 3.2.1. College Alcohol Drinking

The available alcohol drinking data represent prevalence (proportion of cases at a certain time) and not incidence (new cases over time period). This is because the data are based on the survey where the drinking pattern estimates are obtained by asking individuals their drinking behavior during past one year. Hence, data can be interpreted as the number of individuals in certain drinking category at a particular time. Therefore, we use formula given in Equation (Equation 5) to estimate β(t). We assume that drinking is a result of social influences exerted by drinkers (*I*) on susceptibles (*S*) or social drinkers. Individuals recovered from drinking at a constant rate α (i.e., α(t)=α). The recovery is assumed to be permanent (i.e., γ(t)=0). The incoming and departure rates are same (i.e., μ(t)=b(t)=μ) and p=1. These assumptions are reasonable in context of the type of data (college population) used here.

Alcohol drinking data, obtained from Engs et al., 1997 and 1999, is given in the Table A4 [12,13] that represent the trend observed in national college drinking surveys. The recovery rate, α is taken to be 0.17 [4]. We estimate β(t) using simplified Equation (Equation 5) and above assumptions as follows
(20)$$\beta \left(t\right)\approx \frac{\frac{I(t)-I(t-h)}{hI(t)}+\alpha +\mu -\frac{\bar{p}\mu }{I(t)}}{\left(S\left(0\right)+I\left(0\right)\right)e^{-\mu t}-I\left(t\right)+e^{-\mu t}hc_{1}}$$
where
$$c_{1}=\frac{f_{2}\left(0\right)+f_{2}\left(t\right)}{2}+\sum_{k=1}^{n-1}f_{1}\left(kh\right),$$
and
$$f_{2}\left(x\right)=e^{\mu x}\left[\mu -\alpha I\left(x\right)\right].$$

If μ=0, this equation can be reduced, where $f_{2}$ is −αI(x).

We found that mean estimate of β is 1.04 (std = 0.3; Table A4 see Appendix A and Figure 8) during 1982–1994 for the national college drinkers. The estimates of β are comparable to the estimates obtained in the [4]. These estimates of β(t) are all contained in 95% CI of the estimates in the [4], which are $\beta_{0}=1.69$ (95% CI [0.63, 2.75]) and $\beta_{2}=0.75$ (95% CI [0.29, 1.21]).

Engs et al., 1994 and 1997 suggest that 65% of freshmen are drinkers during the start of Fall semester. Hence, we assumed that 0.65 proportion of incoming students are drinkers, i.e., p=0.35. We assumed negligible change in size of a college population and consider rate of enrollments equal to combined rate of graduation and dropout rates (i.e., b(t)=μ(t)=μ).

#### 3.2.2. Obesity Epidemic in US

We use model to see whether weight gain in one person is associated with weight gain in his or her family members and friends. Obese persons are an individual whose body-mass index (the weight in kilograms divided by the square of the height in meters) is greater than or equal to 30. It is found that there has been increasing number of obese persons in a community and a person’s chances of becoming obese increases dramatically if he or she had a parent, sibling, friend or spouse who became obese in a given interval [24]. The most reasonable explanations for the obesity epidemic, include changes in which luxuries and food consumption are being promoted in the society and has not spared any socioeconomic class. An obesity is a result of individual’s choice and behavior which is influenced by appearance and behavior of others in the community. Hence, it suggests that just as with the spread of drug-use or infectious diseases, weight gain in one person might influence weight gain in other person, i.e., it is not that obese or non-obese people simply find other similar people to hang out with. This influence could be direct or indirect, which can vary continuously over time and may depend on demographic and social factors of the community as well.

We used annual CDC data from references [14,18] to estimate parameters for our obese epidemic model. The data obtained from [25] include an age-adjusted prevalence of obesity in US using the projected 2000 U.S. population. The model assumes constant population and hence b(t)=μ(t)=μ. It is assumed that 6% of children are born obese [14]. The value of recovery rate is assumed to be equal to an average of rate at which an overweight individuals move on diet ($4.068\times 10^{-3}$ per week [18]) and rate at which an obese individual stops or reduces bakery, fried meals and soft drinks consumption ($4.4379\times 10^{-3}$ per week [18]). We assume obesity reduces life span by 6 to 7 years. Hence if average life span in US is 78.4 years than average life span of at-risk population for obese is (78.4−6.5) years. The estimated β from [18] ranges from 0.02 to 0.04. These estimates are much lower than our estimated values in Table A5 (see Appendix A) with range of (0.36, 3.02) (Figure 9). This is because the region of our study differs from the region modeled by [18]. Our results suggest that estimates of transmission coefficient increase with increase in μ and decrease in initial size of susceptible population, S(0).
(21)$$\beta \left(t\right)\approx \frac{\frac{I(t)-I(t-h)}{hI(t)}+\alpha +\mu -\frac{\bar{p}\mu }{I(t)}}{\left(S\left(0\right)+I\left(0\right)\right)e^{-(\gamma +\mu )t}-I\left(t\right)+e^{-(\gamma +\mu )t}hc_{2}}$$
where
$$c_{2}=\frac{f_{3}\left(0\right)+f_{3}\left(t\right)}{2}+\sum_{k=1}^{n-1}f_{3}\left(kh\right)$$
and
$$f_{3}\left(x\right)=e^{(\gamma +\mu )x}\left[\gamma +\mu -\alpha I\left(x\right)\right].$$

### 3.3. Estimation of Time-Dependent Transmission Coefficient Using Synthetic Prevalence and Incidence Data

We demonstrate our method of using prevalence and incidence data to estimate time-dependent transmission coefficient using synthetic prevalence and incidence data generated with two choices of transmission coefficients and the model ((Equation 1) and (Equation 2)) with rest of parameters given by (Table 5).

In the first case, we assumed transmission coefficient to be constant over time and in the second case, we consider a transmission coefficient that is seasonal. They are given by

1.β(t)=20,2.β(t)=20(1−ϵcos2πt), with ϵ=0.1.

We generate daily prevalence and incidence data for two years and estimated monthly transmission coefficient using Equations (Equation 5) and (Equation 8) respectively. We used MATLAB’s ’pchip’ function to interpolate the synthetic prevalence and incidence data in the formulation and integrated using ’integral’ function. The monthly estimates for time-dependent transmission coefficient were reasonably accurate and close to the true values of the transmission coefficients used to generate prevalence and incidence data in both the cases when transmission coefficient was constant and when it was periodic (Figure 10). As birth rates, mortality rates and recovery rates are often considered constant in models, we used constant terms for these variables to limit simulation time. However, if time-dependent information on these variables is available they can easily be incorporated and simulated.

## 4. Discussion

Compartmental models have provided valuable insights into the epidemiology of many infectious diseases. Transmission coefficient, a product of contact rate and probability of transmission given a contact, is a parameter in the compartmental model which naturally varies over time. This coefficient had the greatest effect on predictions of dynamics of disease or social problem and difficult to estimate. However, due to lack of detailed data as well as complexities involved in numerical estimating this parameter, most studies estimate it as a time-independent parameter averaging it over the course of epidemic. In this study, we present a method to estimate time-dependent transmission rate using two types of data commonly reported during infectious disease outbreaks: the time series of the number of infectives (or prevalence) and the number of new cases generated during a period of time (or incidence). By deriving an analytical method that uses a standard deterministic model and these data sets to directly estimate β(t), this new approach resolves the computational challenges often involved with more complex model. By applying our approach to several infectious diseases, we illustrate applicability of our methods in various contexts. Moreover, similar approaches can be applied with any appropriate mathematical model to derive time-dependent transmission rate for diseases whose dynamics may need to incorporate other factors such as environment (for. e.g., role of waterbodies in cholera spread) or vector dynamics (for. e.g., impact of mosquito in dengue transmission).

Utility of approach presented in this manuscript is demonstrated using several public-health problems including Ebola, Visceral Leishmaniasis, US college alcohol drinking and obesity outbreak in the US. In particular, we estimated the temporal estimates of transmission rate for Ebola during 2014–2016 outbreak in West Africa (aggregated) as well as for individual countries of Liberia, Guinea and Sierra Leone. Our results though limited by the accuracy of data, demonstrated the wide-variability in transmission risks across the three countries. Moreover, we found that our temporal estimates of transmission risk followed the pattern of incidence closely, but slightly delayed, reflecting the substantial contribution of transmission risk towards the nature of disease progression. During the times of public-health emergencies due to an infectious disease outbreaks such as Ebola outbreak in West Africa or ongoing COVID-19 pandemic, effective reproductive numbers are often estimated using incidence data to understand the progression of disease and inform strategies to curb the transmission. Although estimates of effective reproductive numbers are useful, combining it with estimation of time-varying transmission risk through our approach can be more informative to inform public-health decision-making. Transmission risk at a particular time is a product of contacts and probability of transmission. Thus, it can be used to make short term predictions about new infections as well as it can inform how much reduction in contact patterns or risk of transmission (through mask/vaccination/hygiene) can reduce the transmission parameter sufficiently to reverse the trend of an epidemic.

In the current study, we used simple deterministic model along with simple integration numerical techniques to show how commonly reported data (incidence and prevalence) can be used in informing temporal transmission risk, and thus manage public-health challenges more effectively. Practical application of our approach would improve with use of more complex models (appropriate) as well more sophisticated integration techniques. Moreover, analytical derivation can be used to understand the impact of changes in any other input parameter (such as smaller/longer quarantine periods) on transmission risk in a straight-forward way. Similarly, an area of future research can expand presented framework to understand how incomplete data may alter the quality of parameter estimation. Therefore, value of analysis reported here is as a beginning point for future research that will build on current approach to develop computational models that can inform policies in swift manner during public-health emergencies. We believe using our methods can provide good approximation of time-dependent transmission coefficients and goodness of approximation should increase with use of more sophisticated modeling techniques.

## Figures and Tables

**Figure 1 tropicalmed-06-00141-f001:**
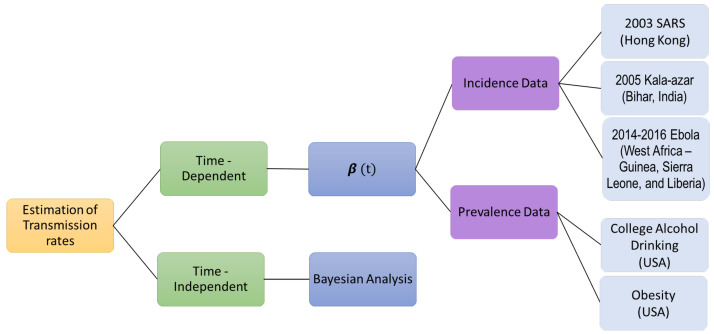
Overview of the paper.

**Figure 2 tropicalmed-06-00141-f002:**
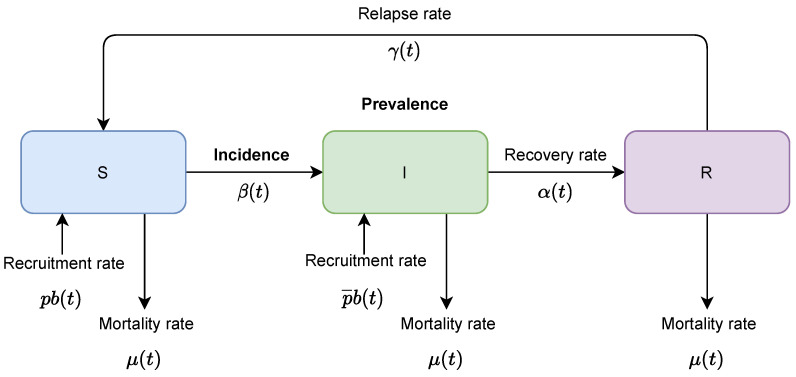
Schematic diagram for the SIR model.

**Figure 3 tropicalmed-06-00141-f003:**
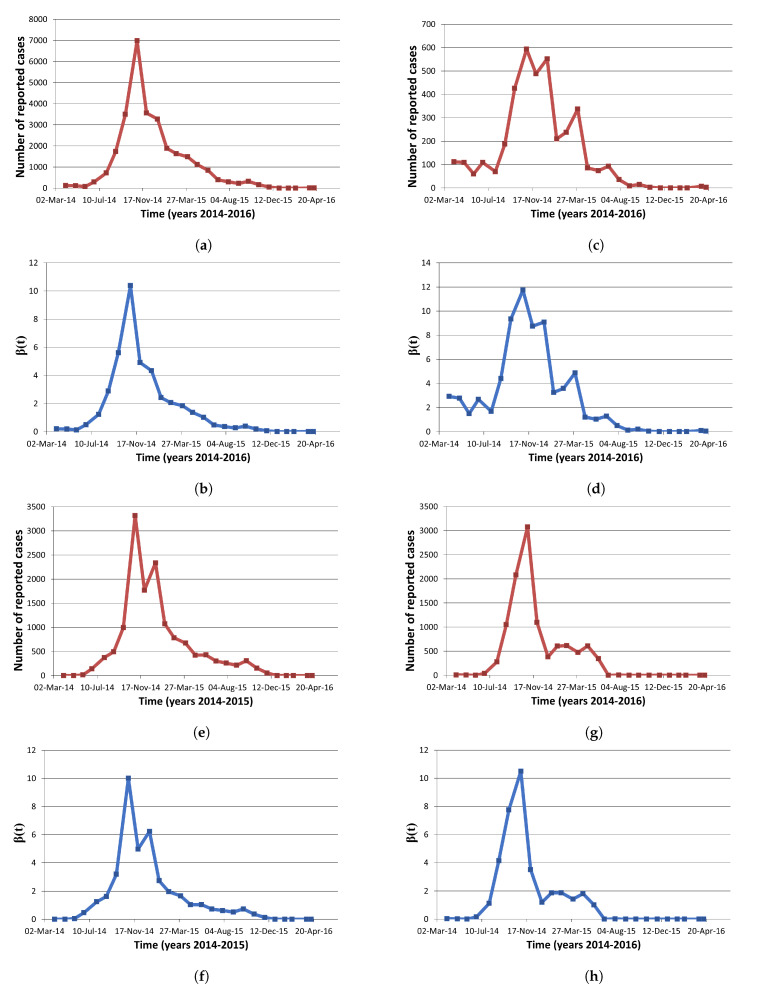
*β*(*t*) estimates for the 2014–2016 Ebola outbreak: (**a**) Cases per month for West Africa; (**b**) *β*(*t*) estimates for West
Africa; (**c**) Cases per month for Guinea; (**d**) *β*(*t*) estimates for Guinea; (**e**) Cases per month for Sierra Leone; (**f**) *β*(*t*) estimates
for Sierra Leone; (**g**) Cases per month for Liberia; (**h**) *β*(*t*) estimates for Liberia.

**Figure 4 tropicalmed-06-00141-f004:**
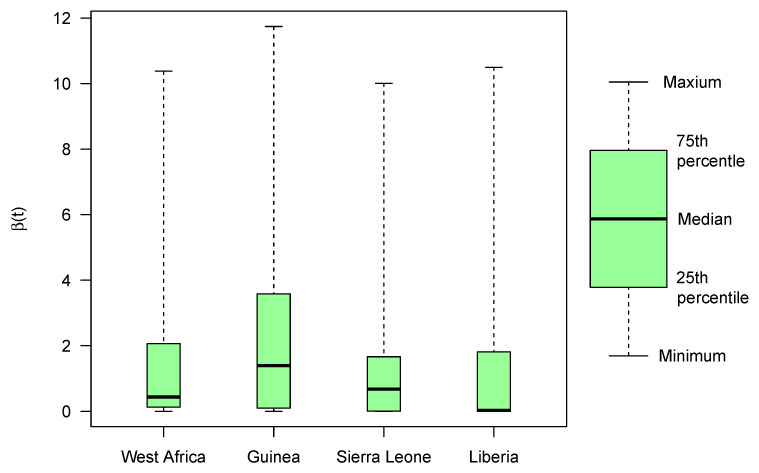
Box-and-Whiskers plot for estimates of β(t) using available incidence data for the 2014–2015 Ebola epidemic of West Africa.

**Figure 5 tropicalmed-06-00141-f005:**
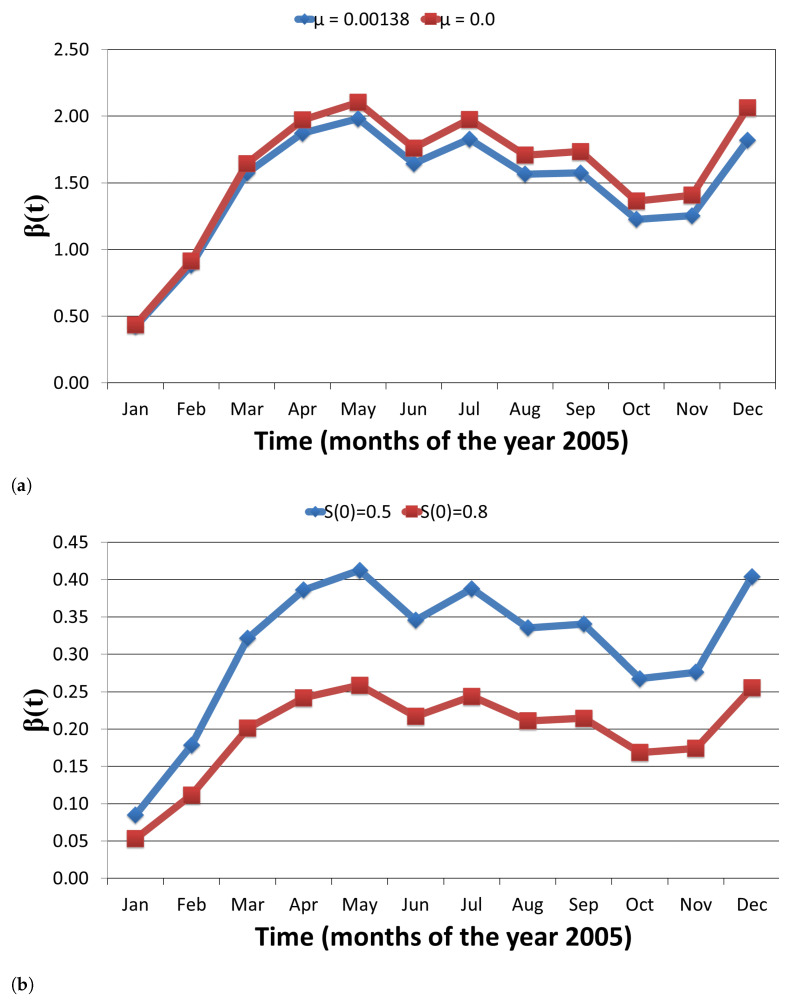
Estimates of *β*(*t*) for the 2005 outbreak of Visceral Leishmaniasis in the Indian state of Bihar,
using available incidence data: (**a**) Estimates of *β*(*t*) related to an outbreak of Visceral Leishmaniasis,
when the initial value of susceptibles, *S*(0) = 0.1; (**b**) Estimates of *β*(*t*) for two initial proportion of
susceptibles in a population affected with Visceral Leishmaniasis. Estimates of *β*(*t*) obtained for two
different values of the mortality rate, *μ*, (0.0 and 0.00138) are almost same.

**Figure 6 tropicalmed-06-00141-f006:**
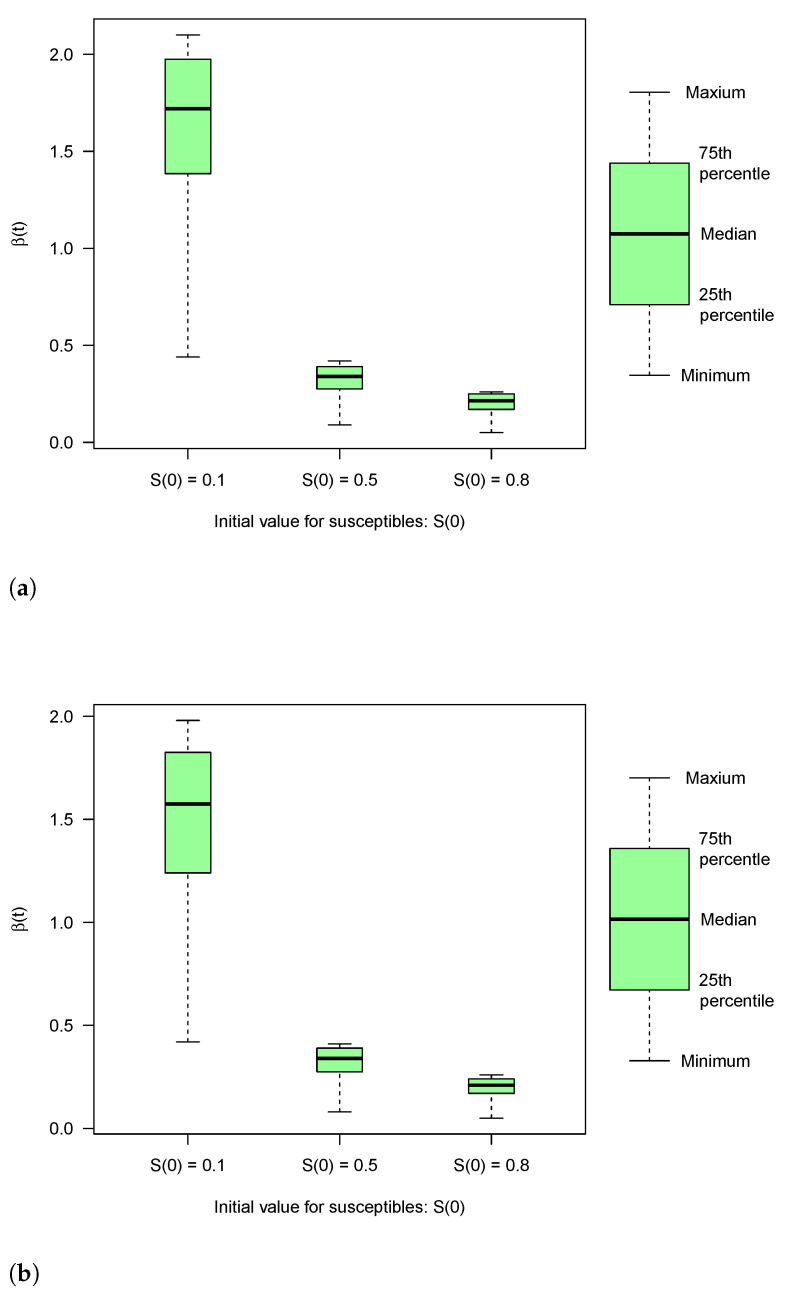
Box-and-Whiskers plot for estimates of *β*(*t*) for two different values of *μ* in a Visceral Leishmaniasis outbreak: (**a**) Estimates for *μ* = 0.0; (**b**) Estimates for *μ* = 0.00138.

**Figure 7 tropicalmed-06-00141-f007:**
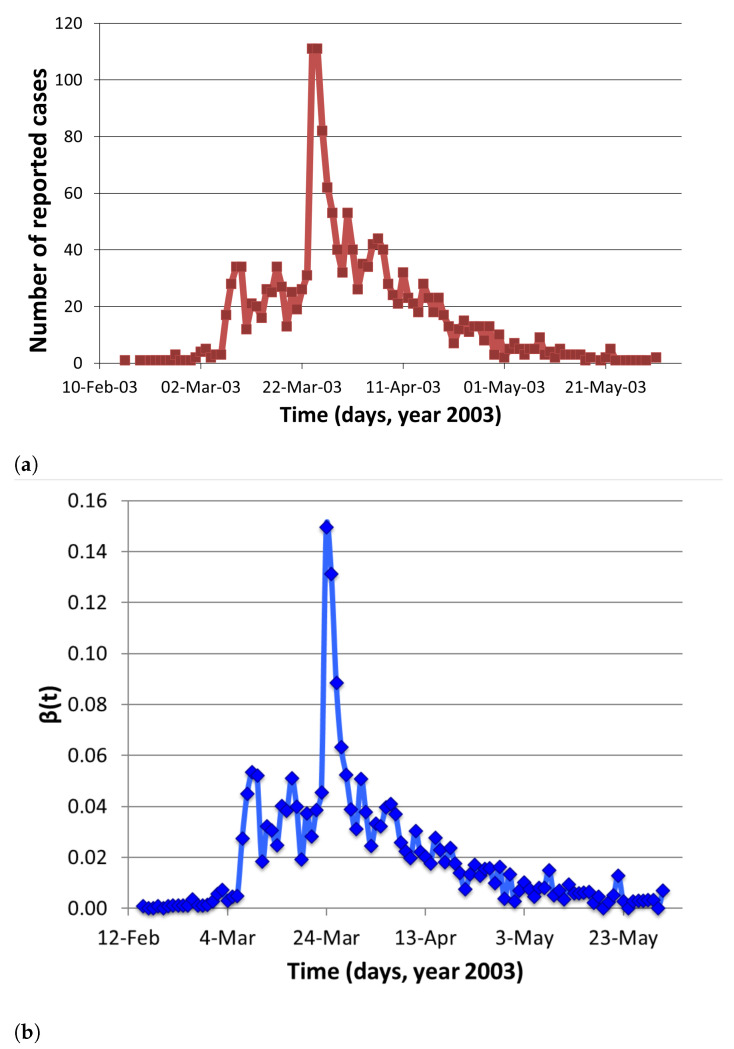
The 2002–2003 SARS outbreak in Hong Kong: (**a**) Daily reported cases; (**b**) Estimation of *β*(*t*) using available incidence data. Prevalence for 16 April 2003 was taken in the calculation when number of symptomatic cases started declining [23].

**Figure 8 tropicalmed-06-00141-f008:**
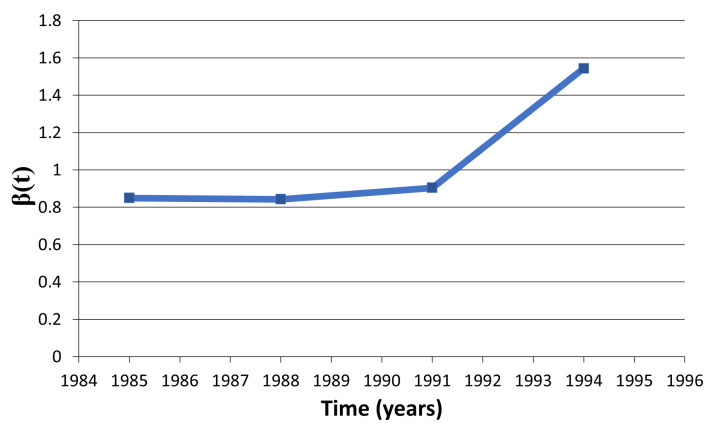
Estimates of β(t) related to alcohol drinking college population, when μ=0.

**Figure 9 tropicalmed-06-00141-f009:**
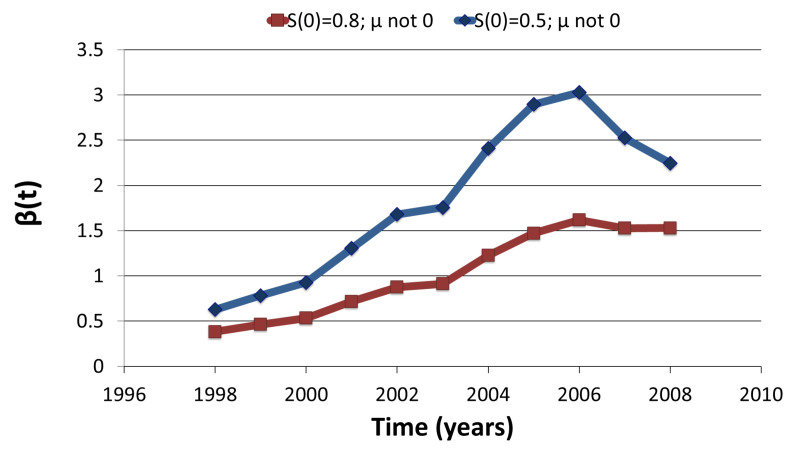
β(t) estimates of Obesity with initial (1997) prevalence of 19.5%.

**Figure 10 tropicalmed-06-00141-f010:**
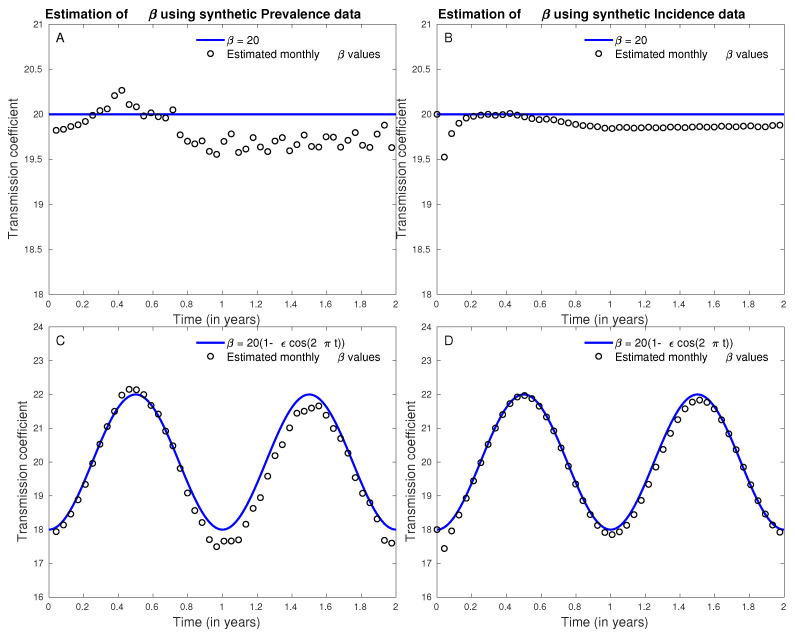
Monthly estimates of transmission coefficients and their true values used to generate synthetic prevalence data (**A**,**C**) and synthetic incidence data (**B**,**D**) for constant transmission coefficient (**A**,**B**) and periodic transmission coefficient (**C**,**D**).

**Table 1 tropicalmed-06-00141-t001:** Some common differences and similarity in estimation methodology between the current study and that in Pollicott et al. (2012).

	Pollicott et al. (2012) [10]	Mubayi et al. (Current Study)
**Data Modified (fitted to a functional form)**	√	×
**Applicable to Prevalence**	√	√
**Applicable to Incidence**	×	√
**Requires** β(0)	√	×
**Requires S(0)**	×	√
**Multiple Applications**	×	√

**Table 2 tropicalmed-06-00141-t002:** Definition of variables and parameters in the model given by Equations (Equation 1) and (Equation 2).

Variables	Definitions
S	Proportion of susceptibles
I	Proportion of “infected and infectious”
R	Proportion of recovered individuals
**Parameters**	**Definitions**
b(t)	Rate of recruitment in the population
p	Proportion of new recruits that are susceptibles
$\bar{p}=1-p$	Proportion of new recruits that are infectious
β(t)	Transmission or influence coefficient
α(t)	Per-capita recovery rate (its reciprocal is infectious period)
γ(t)	Per-capita rate of losing immunity or relapse rate
μ(t)	Per-capita mortality or departure rate

**Table 3 tropicalmed-06-00141-t003:** Definition of variables and parameters in the model given by Equations (Equation 1) and (Equation 2).

Parameters	——– Estimates ——–				
	Ebola	Alcohol	Obesity	Kala-Azar	SARS
p	1.0	0.35 [12,13]	0.94 [14]	1.0	1.0
	(per month)	(per year)	(per year)	(per month)	(per day)
b	0.0	0.29 [15]	0.01	0.003 [16]	0.0
α	0.003 [17]	0.17 [4]	0.22 [18]	0.211 [7]	0.04 [19]
γ	0.008 [20]	0.0	0.14 [18]	0.0	0.0
μ	0.0	0.27 [4]	0.013	0.001 [7]	0.0

**Table 4 tropicalmed-06-00141-t004:** Summary statistics of Ebola results for β(t).

	Minimum	Mean±SD	25th Percentile	Median	75th Percentile	Maximum
West Africa	0.00	1.57±2.36	0.14	0.43	2.01	10.38
Guinea	0.00	2.73±3.34	0.10	1.36	3.50	11.74
Sierra Leone	0.00	1.51±2.30	0.02	0.68	1.65	6.23
Liberia	0.00	1.40±2.52	0.00	0.03	1.71	10.50

**Table 5 tropicalmed-06-00141-t005:** Parameters for generating synthetic prevalence data.

Parameters	*p*	b(t)	γ(t)	μ(t)	α(t)
Values	1	33/1000	0	33/1000	8

## Data Availability

In this section, the data are collected from public domain and is included in the Tables in Appendix A.

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
