# Peer review of "Analytical Estimation of Data-Motivated Time-Dependent Disease Transmission Rate: An Application to Ebola and Selected Public Health Problems"

_tropicalmed, 2021, doi:10.3390/tropicalmed6030141_

Round 1
Reviewer 1 Report
This is a very interesting research of mathematical modeling. The results seem original and new. The paper is well organized. I recommend it for publication as it is.
Author Response
Thank you for your feedback. We appreciate that you did not want us to make any changes. We are making few changes to increase more readability according to the suggestions of other reviewers.
Reviewer 2 Report
Mubayi et al propose a method to estimate time-dependant disease transmission rates and apply it to Ebola, SARS and Leishmaniasis as well as obesity and alcohol drinking.
I have two major problems with this study.
- The authors’ method is only applied to real-world data. To be able to assess the accuracy of the obtained estimates I would strongly suggest to estimate the transmission rates of simulated data with a known underlying transmission rate before applying the method on real-world data. With this approach, the error in the transmission rate estimates can be quantified exactly.
- There are other methods to estimate time-dependant transmission rates. Are these doing better or worse? Comparing them (best on the simulated data) with the authors’ methods will really add value to the manuscript.
Minor comments:
- The authors explain in great detail what prevalence and incidence is which I would expect the informed reader to know that at least partly. But then the methods with all their equations is explained in much less detail. I think this approach would be appropriate for a theoretical journal, but Tropical Medicine and Infectious Disease is not a theoretical journal. So I would suggest to explain your methods and also the results in more detail and in a rather clear, simple language.
- The first paragraph of the Results Section seems to belong more into the Methods Section.
- Table 1.
- “Proportion of recovered individuals” – Are they also immune?
- “Proportion of new recruits that are susceptibles”. With “recruits” you mean births, don’t you? Why not phrase it like that and make it easier for the reader?
- “Proportion of new recruits that are infected” shouldn’t that read “…that are infectious”?
- Figure 2. – Q0? I can’t see Q0 on the graph? I presume Q4 is the max value? Why not order the Qs in the legend from Q0 to Q4?
- Table 3 – Define what Q stands for. I’m slightly confused about Q4 and max value. I would presume Q4 to be the maximum value?
- Figure 4 – What is S(0)? Please define. What is mu? Please define.
- Where is Figure 5?
- Figure 6 – What are the “??”. What is mu? Please define.
Author Response
- Response to Reviewer-2
- The authors’ method is only applied to real-world data. To be able to assess the accuracy of the obtained estimates I would strongly suggest to estimate the transmission rates of simulated data with a known underlying transmission rate before applying the method on real-world data. With this approach, the error in the transmission rate estimates can be quantified exactly.
We have included subsection 3.3 where we discuss estimation of time dependent transmission coefficient using synthetic prevalence data. We demonstrate our method of using prevalence data to estimate time dependent transmission coefficient using synthetic prevalence data generated with two particular choices of transmission coefficients (constant and seasonal with respect to time). This section was previously in the Appendix, now we have included it under Result to increase more visibility for the readers.
- There are other methods to estimate time-dependent transmission rates. Are these doing better or worse? Comparing them (best on the simulated data) with the authors’ methods will really add value to the manuscript.
We have discussed other methods for estimating time-dependent transmission rates in the Introduction. We have also added the theoretical work of Pollicott, Wang et.al which focuses on analytical estimation of time-dependent transmission rates on incidence data, whereas our method focuses on prevalence and incidence data both as well as we focus on applying our expressions to multiple public health problems. We have added a comparison table in the manuscript and an added text is included in page 3 of the manuscript.
- The authors explain in great detail what prevalence and incidence is which I would expect the informed reader to know that at least partly. But then the methods with all their equations is explained in much less detail. I think this approach would be appropriate for a theoretical journal, but Tropical Medicine and Infectious Disease is not a theoretical journal. So I would suggest to explain your methods and also the results in more detail and in a rather clear, simple language.
We have made attempts to explain our methods in details, added a diagram to our SIR model. We also have added a table in the result section to better explain the reach of our estimation.
- The first paragraph of the Results Section seems to belong more into the Methods Section.
The first paragraph in Result section gives the first order discretization for derivatives to compute estimates of β(t) which we have used in the examples and hence it is in the Result section.
- Table 1.
- “Proportion of recovered individuals” – Are they also immune?
Yes, we have now provided a diagram for the mentioned model and have also discussed our assumption for the model. The recovered individuals are immune.
- “Proportion of new recruits that are susceptibles”. With “recruits” you mean births, don’t you? Why not phrase it like that and make it easier for the reader?
Since our parameter captures immigration, emigration as well as birth rate, we therefore use recruitment rate.
- “Proportion of new recruits that are infected” shouldn’t that read “…that are infectious”?
We have corrected it in our Table and it now reads as infected and infectious.
- Figure 2. – Q0? I can’t see Q0 on the graph? I presume Q4 is the max value? Why not order the Qs in the legend from Q0 to Q4?
We have fixed the figures and have defined the quartiles as Q. We have discussed this on our paper.
- Table 3 – Define what Q stands for. I’m slightly confused about Q4 and max value. I would presume Q4 to be the maximum value?
We have defined the quartiles as Q and have mentioned them in tables and texts in the manuscript.
- Figure 4 – What is S(0)? Please define. What is mu? Please define.
We have added the initial value of susceptibles and the mortality rate mu in Fig. 5(previously Fig. 4 )
- Where is Figure 5?
We have added the required figure.
- Figure 6 – What are the “??”. What is mu? Please define.
We have fixed ?? it and mu as the mortality rate, which we have added in the text.
Reviewer 3 Report
Refer to attached file.

Author Response
- Response to Reviewer 3
- Published work using an approach which is similar to the paper should be discussed and compared with the formula derived by the authors. For example, the paper by Pollicot et al. (2012) presented formulae derived from the system of equations of the SIR model.
We have discussed the two different approaches in our introduction and have added a table of comparison for the two studies, see Table 1, page 3.
- Minor comments 1. Page 4, equations 1 and 2: Define S/ (dS/dt). 2.Page 5, Line 27: bayesian should be Bayesian. Line 30: “If there are more than one data sets are used ”. Rephrase as “If more than one data sets are used”.
We have addressed these issues in the manuscript.
Round 2
Reviewer 2 Report
I thank the authors for their changes and I think that the paper has already benefited. However, as already mentioned in the first review round (second comment), I still think it could benefit immensely from a comparison (a quantitative one) with other methods. It is great to mention at least one method in the introduction, but it still does not tell the reader how good the new method performs compared to already existing methods. Therefore, I would like to strongly suggest to add a real comparison with other methods, not just discussing them.
Author Response
We thank the reviewer for suggesting a quantitative comparison of our methods with other existing methods in the literature. However, due to different focus and goals of our manuscript, we consider the quantitative approach for our future studies. The goal of the current manuscript is to derive a novel estimation method using analytical approach that can use sparce data such as those of incidence and prevalence for a new or emergent disease and provide estimates of transmission rate, a key quantity for transmission dynamics. Further, we also focus on application of our method to different public health problems. Hence, quantitative comparison of our method with other methods in literature is not our current focus. However, at this point it is worth mentioning some more details of the estimation method by Pollicott et.al and also highlight how it differs from our method. A substantial difference exists between ours and Pollicott et.al. as the later has several limitations unlike ours.
First, the proportion of infected individuals, f(t), cannot decrease too fast over the [full] time interval [of interest]. In general, one can add a sufficiently large constant to f(t) to ensure [this] fast rate of decline in number of cases, but this will change the range of [applicable] reasonable β(0). Furthermore, applicability of β(0) needs to be checked based on prevalence data [case by case].
Second, one must assume that the proportion (or number) of notifications is always strictly positive, a strong assumption. [In practice], some of these restrictions [can] may be overcome by replacing zero reporting in the time series data with a very small positive value.
Third, for a chosen β(0), the algorithm may only apply to a finite length of infection data.
Finally, one either needs to know the value of the transmission rate at some fixed time, or verify that the claimed properties of β(t) hold for all β(0) attaining a large range of values.
In contrast to this, our method is an analytical method that does not need fitting of data to spline or predefined functions, we use the data and model to directly estimate β(t). We also did provide a table in the paper citing the differences and the similarities to understand the methods better.
Reviewer 3 Report
The authors have adequately addressed the points raised.
Author Response
Thank you for helping us make the manuscript better with the improvisations.